# Best-Worst Scaling Survey of Inpatients’ Preferences in Medical Decision-Making Participation in China

**DOI:** 10.3390/healthcare11030323

**Published:** 2023-01-21

**Authors:** Tao Sun, Hanlin Chen, Yuan Gao, Yingru Xiang, Feng Wang, Ziling Ni, Xiaohe Wang, Xianhong Huang

**Affiliations:** 1Department of Health Policy and Management, School of Public Health, Hangzhou Normal University, Hangzhou 311121, China; 2The Affiliated Hangzhou First People’s Hospital, Zhejiang University School of Medicine, Hangzhou 310006, China

**Keywords:** best-worst scaling, preference, patient participation, shared decision-making, inpatient

## Abstract

This study assesses inpatients’ preferences for participating in medical decision-making and determines the factors’ rankings in order of importance and whether they vary for respondents with different characteristics. Case 1 best-worst scaling (BWS) was used for the study design. Thirteen attributes influencing inpatient medical decision-making participation were identified based on a literature review and interview results. A balanced incomplete block design was used to form choice sets for the BWS questionnaire for a cross-sectional study examining inpatients’ preferences for participating in medical decision-making. Based on results from 814 inpatient participants, the three most important factors influencing inpatients’ medical decision-making participation were inpatients’ trust in physicians, physicians’ professional expertise, and physicians’ attitudes. The mixed logit model results reflect the significant heterogeneity in respondents’ preferences for shared decision-making. To facilitate resource allocation, improve the physician-patient relationship, and encourage patient decision-making participation more actively and effectively, decision-makers should emphasize patients’ trust, enhance physicians’ ability to diagnose and treat diseases, and improve their attitudes toward providing care and communication from the perspectives of patients, physicians, and the social environment. Further research is needed on the heterogeneity of patients’ preferences for participating in medical decision-making and how to improve patient participation.

## 1. Introduction

“Patient-centeredness” is increasingly important in modern healthcare [1]. Patients’ medical decision-making power has increased alongside the development of the healthcare industry, improvement in medical models, and an increase in health needs [2]. With the popularization of medical knowledge and increased cultural awareness [3], most patients can comprehend medical knowledge and participate in decision-making—this paves the way for the shared decision-making model (SDM) [4]. At the heart of SDM is the promotion of patient participation in the healthcare decision-making process. Patient medical decision-making participation is a proactive treatment step [5], wherein patients and relatives discuss the illness and treatment options and reach a consensus.

### 1.1. Thematic Background

The importance of patients’ participation in medical decision-making has been recognized worldwide in healthcare. Studies have shown that SDM can better combine patients’ willingness, values, and choices with treatment options and clinical evidence, and further integrate the patient-centered medical model with evidence-based medicine [6]. Research shows that patient medical decision-making participation improves their experience, compliance and satisfaction [7,8,9,10], and cognition of physician-patient relationships and communication [11]. It also contributes to decision-making quality, patient’s illness cognition and medical risk perception [12], and reduced misuse of antibiotics [13], thus accelerating the development of personalized treatment and precision medicine. There have been studies on factors influencing patient participation in medical decision-making. According to the literature review, these factors can be divided into three levels: patient, physician, and social environment [10]. Patient characteristics (e.g., age, education level, gender) and factors related to decision-making (disease characteristics, therapeutic options, and availability of scientific evidence for treatment efficacy) potentially influence patient participation [4,14,15]. Other elements include patients’ trust in physicians, satisfaction, and ability to participate in decision-making [16,17]. From the physician’s perspective, factors influencing such patient participation include the consultation time duration [16], work attitudes [18,19], and communication ability regarding decision-making [20,21]. Socioeconomic factors may influence inpatients’ cognition regarding medical decision-making involvement [22]. With the advancement of medicine, patients have the option of choosing from more than one treatment plan for one disease, thus allowing them to exercise their own preferences and value judgments. Patient preference information helps to not only tailor clinical interventions but also guide clinical decision-making when there is no consensus on the preferred solution for a health problem. Many studies have discovered that consonance of patient preference for decision-making and the decision made by physicians can lead to better patient satisfaction with the decision and better patient mental health [23]. Gao et al. recently found that patient preference for decision-making influences decision-making quality, which in turn affects patient satisfaction. This means that patient preference for decision-making is a key predictor of decision-making quality [24]. However, under the influence of traditional Chinese culture, patients’ family members usually play a pivotal role in decision-making; they not only affect patients’ medical decision-making but sometimes also make decisions in their stead [25]. Further, when patients have low level of awareness regarding their rights, they tend to rely on physicians’ authority too much. This means that Chinese patients’ ability to participate in medical decision-making and the ratio of those who do so are yet to be improved [26]. For example, based on recent research by Xiao et al. [27], cancer patients have a low to medium level of shared decision-making—much lower than the level in the study by Hahlweg et al. [28]—and a low level of participation in treatment/nursing decision-making, thus implying unsatisfied needs for participation. Likewise, Li’s study [29] documented that 36.5% of patients with primary liver cancer have decisions made by their family members. At the same time, half of them do not think their family members engage in decision-making to the same extent as they want them to [29]. Therefore, it is necessary to explore patient preference for and factors influencing their participation in medical decision-making in the Chinese background.

### 1.2. Methodological Background

As mentioned above, the factors influencing patient involvement in healthcare decision-making behavior have been explored in the academic community. However, no study has addressed the relative importance of these factors. In terms of research methodology, the Likert scale is often used to measure relative importance [30], while choice experiments are used less frequently. This is particularly important to identify treatment and care options that are more cost-effective and in line with individuals’ desires [31,32]. The National Institute for Health and Care Excellence Committee, a British health technology assessment body, has accelerated patient preference data use in one of three preference-related priority areas [33]. These experiments accurately estimated participants’ preferences in an efficient, highly cost-effective, and generalizable manner [34,35,36,37]. Best-worst scaling (BWS) is a method based on random utility theory and a multi-criteria decision-making (MCDM) process (i.e., a decision-making process of choosing exclusively from conflicting options). BWS is a novel method of data collection for measuring willingness and preferences, and an ideal choice experiment for eliciting priorities [38,39], because it can adequately assess the relative importance of factors for patients when there are multiple factors involved [40]. It is a more explicit, accurate, and reliable method for measuring preferences and willingness [41]. There have been empirical investigations that have employed BWS for data collection and decision analyses using MCDM. Research shows that BWS can help decision-makers select the best option with limited health service resources [42]. Kaya Pezük et al. explored the rankings of COVID-19 vaccines’ side effects with a soft decision-making approach and summarized the strengths and weaknesses of MCDM methods [43]. Turbitt established parents’ priorities when deciding whether to let children with fragile X syndrome participate in clinical drug trials [44]. Paquin et al. determined rankings of factors affecting health workers’ and patients’ participation in early-phase gene therapy trials for Duchenne muscular dystrophy with the BWS method [45]. Based on the above examples, BWS is feasible and acceptable in domestic and overseas research, although, compared with discrete choice experiments, it is not sufficiently mature in terms of providing explanations [46,47,48]. Research has been conducted on patient decision-making using the BWS [49]. However, the attributes and factors that inpatients value when participating in medical decision-making need to be explored, and research in the Chinese context is still lacking. The Chinese culture, to some extent, hinders patient involvement in treatment decision-making. It is especially critical for inpatients with more severe diseases to participate in medical decision-making. However, factors influencing patient participation in decision-making and their relative importance are not apparent yet. To facilitate patient involvement in treatment decision-making, health workers must first understand how patients make clinical decisions. Considering the status quo, it is necessary to explore more factors influencing patient involvement in medical decision-making with the BWS method and examine inpatient preferences in the decision-making process in the Chinese context.

Therefore, this study aimed to include a more comprehensive set of attributes and a wider range of respondents compared to previous studies. To this end, we used a quantitative technique, the BWS (Case 1), to investigate the relative importance of factors affecting inpatients’ medical decision-making participation to explore preferences in Zhejiang Province. By doing so, we aimed to establish the rankings of predictors of patient involvement in clinical decision-making with a list of critical factors. In addition, the purpose of this study was to determine if these factors’ influences are identical among respondents with different characteristics, such as inpatient departments and age. This method produces valuable measures to better apply the “patient-centeredness” concept, promote patient-physician interaction, help patients participate in decision-making more actively and effectively, and increase patients’ decision-making agency to ensure that patients receive desired treatments. Simultaneously, the study findings can help develop relevant educational campaigns and facilitate patient-centered decision-making.

## 2. Methods

### 2.1. Best-Worst Scaling Experiment

BWS is based on the hypothesis that all (health-related) products or services can be described according to characteristics (attributes or levels) [50]. In various scenarios imitating real-life decision-making scenarios, participants are asked to repeatedly choose their preferred option from two or more alternatives based on several attributes, where each attribute level varies across alternatives [51]. The relative importance of each attribute indicates which attribute most affects the participant [39,51,52]. Researchers can then make informed suggestions by analyzing whether each attribute level has a statistically significant influence on decision-making. BWS is a data collection method based on the random utility theory [53]. This theory assumes that one can assess respondent preference for different options by statistically analyzing their behavior of choosing from multiple preference questions. Based on this idea, the BWS approach evaluates rankings of respondent preference by asking them to choose the best and worst options in various combinations, repeatedly. This way, the BWS method helps researchers conveniently collect more data compared to discrete choice experiments (DCE). Moreover, the repetitive nature of data collection helps to establish complete, accurate, and reliable rankings of factors’ importance [41]. When compared to rating scales, including the Likert scale and 1–10 rating scale, BWS can significantly reduce acquiescence bias (agreement bias), social desirability bias (tendency to lie), and extreme response bias in rating scales. It is considered to provide clearer differentiation between variables and is more efficient in the calculation of the importance of individual items as compared to rating scales [54]. Cohen et al. [55] found that when assessing IT managers’ preferences for file server selection, BWS had the highest mean t-value when the *t*-tests were used to compare attribute differences, thus demonstrating that BWS was efficient in performing a clearer differentiation between attributes. Jaeger et al. [56] used a study of consumer preferences for meat pies as an example and verified that BWS made it easier to not only distinguish between the sample’s preferences but also for respondents to answer the questions. Regarding the calculation of the importance of each item, Adamsen [57]—based on a review of the literature—questioned whether the ranking of responses on a rating scale (e.g., the Likert scale, which divides responses into five to seven levels for each item) correctly reflects the importance of each item, and conducted a study on consumers’ preference for organic apples; this study demonstrated that BWS could predict the importance of each item more accurately.

In the existing literature, there are three types of BWS (“cases”), based on differences in the design of the choice set: Case 1 (“object case”), Case 2 (“profile case”), and Case 3 (“multi-profile case”). Case 1 presents items (objects) for respondents to evaluate and construct different subsets from the list using experimental designs. Each subset is presented as a choice set to respondents who are asked to choose the best (or the most important) and worst (or least important) items. The task is repeated several times until all subsets are calculated. Being the easiest of the three, it does not have a level structure and can estimate the general rank of each item in an object list. Unlike Case 1, Cases 2 and 3 present the attributes and their levels. Case 2 BWS studies often use Orthogonal Main Effects Plans to prepare questionnaires. Case 3 presents multiple profiles to individuals; they need to choose the best and worst profiles in each choice set. This study used Case 1 BWS to investigate inpatients’ preferences in medical decision-making.

Intervention studies involving animals or humans, and other studies that require ethical approval, must list the authority that provided approval and the corresponding ethical approval code.

### 2.2. Generation of Best-Worst Scaling Factors

The BWS questionnaire was designed in a four-step process. First, we conducted an extensive literature review. Relevant studies were identified via searches of databases, such as PubMed and Web of Science, and factors were identified after synthesizing and discussing them. Second, we held a focus-group interview to discuss the feasibility of the identified attributes. We selected participants for this focus group using convenience sampling based on inclusion and exclusion criteria. Inclusion criteria were (1) giving informed consent to participate, (2) being aged 18 years or older, (3) being in a stable condition and with good mental status, and (4) being able and willing to participate in this study. Exclusion criteria were (1) incapable and/or unwilling to participate in this study; (2) having a mental illness, difficulty in conversing, or being deaf and mute; or (3) being a patient in a critical care ward. Furthermore, we employed snowball sampling to include 15 inpatients who stayed in the same rooms as the participants. The sample size was determined by reaching data saturation—that is, a point where no more new themes emerged. The purpose and content of the study were explained to the interviewees before the interview, and they were informed of their rights during the interview and that the interview would not interfere with their usual treatment. The interview outline was prepared according to the purpose of the study, mainly including the views and attitudes toward patient participation in medical decision-making, the scope of the participation process, influencing factors, and impact and countermeasure suggestions. After the focus groups, we also conducted expert panel discussions. The authors summarized insights from the literature review and focus group interviews. This content was translated and shared with a multidisciplinary expert panel for discussion and finalization, and factors associated with inpatients’ medical decision-making participation were identified. To evaluate the factors’ feasibility, we invited 30 inpatients to participate in a pilot survey, where they were asked to choose the five most critical factors to them and provide justifications. Finally, the authors reviewed all information and evidence from the previous steps and finalized the BWS questionnaire. The final version comprised 13 attributes (or services). Table 1 presents the description of each attribute.

### 2.3. Questionnaire and Experimental Design

Many Cases 1 BWS studies used a balanced incomplete block design (BIBD) to construct choice sets (e.g., [46,48,67]). BIBD ensures each attribute appears the same number of times as the other attributes and that each pair of attributes appears the same number of times as the other pairs and has features such as being economical, balanced, and flexible [68]. Table 2 illustrates the experimental design with 13 choice tasks, each with four factors. Each choice task provides a brief description of the background and attributes. Figure 1 shows a choice task. For each choice task, the participant chooses two attributes: the most important and the least important. Each participant makes 26 decisions (13 each for the best and the worst). Further information and a detailed guide on choice set construction in BIBDs can be found in Louviere et al.’s study [39].

There is no consensus regarding the optimal sample size for BWS studies. Lancsar and Louviere suggested that over 20 respondents are needed for each choice set [69], but a systematic review shows that the sample size used in Case 1 BWS ranges from 15–803, with a median of 175 [37].

### 2.4. Survey and Data Collection

The primary survey was conducted by five graduate students with adequate in-person survey experience and two research fellows from July 1–September 30, 2020, in Zhejiang, China, which has a population of 65.4 million. In 2019, Zhejiang’s total expenditure on healthcare was 344.053 billion CNY (49.137 billion USD), of which the out-of-pocket cost was 177.719 billion CNY (25.296 billion USD), and the health expenditure per capita of urban residents was 2300.0 CNY (328.52 USD). Using stratified random sampling, we divided the various regions of Zhejiang into four levels: (1) the cities of Hangzhou and Ningbo, which reported GDPs of over 1 trillion CNY (142.817 billion USD); (2) Wenzhou, Shaoxing, and Jiaxing, which reported GDPs ranging from 600 billion to 1 trillion yuan; (3) Taizhou, Jinhua, and Huzhou, which reported GDPs ranging from 300–600 billion CNY (42.84–85.69 billion USD); and (4) Quzhou, Lishui, and Zhoushan, which reported GDPs under 300 billion yuan. We then selected one city from each level: Hangzhou, Jiaxing, Huzhou, and Quzhou. We randomly selected 10 tertiary and secondary hospitals, each, at different levels in the four cities. Convenience sampling was used to collect samples from different departments, including internal medicine, surgery, gynecology, otolaryngology, and orthopedics. According to the size of the hospital, 60 participants were selected from each tertiary hospital, and 30 from each secondary hospital. In total, 900 inpatients were surveyed. Questionnaires with incomplete answers, contradictory answers, or a short completion time were deemed invalid. As a result, of the 865 questionnaires that were completed (a response rate of 96.1%), only 814 (94.1%) were included in the analysis. Table 3 presents the participants’ demographic characteristics.

Surveyors were trained to adopt uniform standards. A one-on-one survey was also conducted. Informed consent was obtained before the survey. Afterward, the questionnaires were reviewed to determine usability. To reduce errors, the questionnaires were coded and entered using double data entries. The questionnaire consists of two sections: (1) the BWS questionnaire prepared based on the steps described in the experimental design, and (2) a series of questions about respondents’ demographics, such as gender, age, education, and income.

### 2.5. Statistical Analysis

The two methods described below were used to analyze the collected BWS data. A counting analysis was used to determine the number of times each attribute was chosen. This includes the following three types of best-worst (BW) scores:The BW score is the number of times an attribute is selected as the most important minus the number of times it is chosen as the least important. If the BW score is a positive number, the attribute is selected as the most important more often than the least important, or vice versa [70].Scaled BW score is the square root of the total best score divided by the total worst score. It designates the choice probability relative to the most essential attribute [71].The mean BW score equals the BW score divided by the number of respondents responding to each attribute.

Based on the Maxdiff model, the function mlogit() was used to fit the conditional or mixed logit (MXL) model to the results of Case 1 BWS choice sets to measure the preferences and heterogeneity of patients’ decision-making participation [72]. Each task provides two modeling results: the best and worst. The BWS analysis is based on random utility theory [54,73]. Specifically, the analyses of the best and worst choices are based on the maximization of utility and negative utility, respectively. Thus, when an attribute was selected as the best or worst alternative, we used virtual code (1) or negative virtual code (−1) to describe the probability of its appearance in a specific combination of attributes [74]. The equation $u_{diff}^{i}$ on the latent utility scale shows the relationship between the BW utility difference of a choice task i (*i* = 1, 2, 3, …, 13) and the 13 independent variables (factors).

Under our model settings, the systematic component of the utility function is
(1)$$v=\beta_{1}environment+\beta_{2}communication+\beta_{3}law+\beta_{4}trust+\beta_{5}expertise+\;\beta_{6}people+\beta_{7}awareness+\beta_{8}time+\beta_{9}ability+\beta_{10}rule+\beta_{11}attitude+\;\beta_{12}literacy+\beta_{13}burden,$$
where *environment*, *communication*, *law*, *trust*, *expertise*, *people*, *awareness*, *time*, *ability*, *rule*, *attitude*, *literacy*, and *burden* are item variables and *β*s are coefficients to be estimated.

Conditional logit analysis was used to estimate the attribute coefficient, which indicates the importance of an attribute relative to other attributes [74]. In the Maxdiff model, the probability of choosing attribute (i) as the best and attribute (j) as the worst from the choice set © can be estimated using the conditional logit model and the systematic component of the utility as follows:(2)$$Pr\left(best=i,worst=j\right)=\frac{\exp\left(v_{i}-v_{j}\right)}{\sum_{p,q∁C,p\neq q}\exp\left(v_{p}-v_{q}\right)}.$$

McFadden’s R-squared (rho-squared) [36] was used to measure model fit. Rho-squared values between 0.2 and 0.4 represent a very good model fit. If the original model does not show a good fit, it may be because heterogeneity was not considered [52]. The MXL model assumes the variables are relative to individuals and therefore considers heterogeneity [69]. We implemented an MXL model for the data to obtain calibrated rho-squared values and coefficients. A one-way analysis of variance was used to analyze heterogeneity. Furthermore, differences in the BW scores among the various subgroups were assessed. Four analyses were performed based on the department (internal medicine, surgery, gynecology, otolaryngology, and others), monthly family income (<10,000, 10,000 to 20,000, >20,000 CNY), number of hospitalizations in the last year (0, 1 or 2, ≥3), and age (≤25, 26 to 35, 36 to 45, 46 to 55, ≥56). Tamhane and Dunnett T3 were used to conduct posthoc tests. R by the R Foundation [75] was used to design the BWS questionnaire and perform all statistical tests, with a significance level of 0.05.

## 3. Results

### 3.1. Respondents’ Demographic Characteristics

Of the 814 inpatients surveyed, the gender ratio was found to be generally balanced, with slightly more females (50.4%). The majority of participants had only completed middle school and below (57.4%); some were under 25 years of age (9.5%), while the other age groups contained about the same number of people. Many were in the surgery department (46.4%), the majority had not been hospitalized in the last year (64.5%), and most had a monthly family income under 10,000 CNY (1.428 USD) (38.1%).

### 3.2. Results of the Best-Worst Scaling Survey

Table 4 shows the BWS survey results. Among the factors influencing patients’ medical decision-making participation, “patients’ trust in physicians” was rated as the most important (mean BW score = 1.581), followed by “physicians’ professional expertise” (mean BW score = 1.359) and “physicians’ attitudes” (mean BW score = 1.327), and “patients’ health literacy” was the least important (mean BW score = −0.988). Table 4 also presents the BW, mean BW, and mean std. BW score, scaled BW score, std. scaled BW score, and ranks of the other factors. Figure 2 shows the mean BW score.

To elicit the relative importance of the attributes for individuals, we present the mean and standard deviation of the BW score for each attribute in Figure 3.

### 3.3. Heterogeneity

Conditional and MXL models evaluated the factors’ relative importance. Each attribute’s importance was estimated using “patients’ health literacy,” the least important attribute, as the reference; that is, a base factor with a coefficient of zero. According to the model fit results, all the variables had significant positive coefficients, meaning that the 12 attributes were more important as independent variables compared with “patients’ health literacy.” Among them, “patients’ trust in physicians,” “physicians’ professional expertise,” and “physicians’ attitudes” were the three most important factors, followed by “physicians’ communication ability” and “consultation time duration.” To determine the relative importance of the 13 attributes, we used bws.sp to calculate their shares of preference, producing results consistent with the above rankings (Table 5).

McFadden’s R-squared (rho-squared) of the initial conditional logit model was 0.0739, indicating not a good fit, meaning that inpatients with different characteristics had different preferences over the factors affecting medical decision-making participation; there was significant heterogeneity [52]. Figure 4 illustrates the distribution of the BW scores for each variable. Multimodal preferences were observed for some factors, consistent with McFadden’s R-squared results. Therefore, we applied an MXL model to our data, resulting in a rho-squared value of 0.2771, meaning that the goodness of fit improved, and the model had a good fit.

Table 6 shows, based on the mean BW scores, the results of the heterogeneity analysis for inpatients in subgroups with different demographic characteristics. In the Appendix A, Figure A1, Figure A2, Figure A3 and Figure A4 present, for different subgroups, the BW scores for the attributes showing statistically significant preference heterogeneity. Overall, patients in all the subgroups were most likely to choose *trust* and *literacy* as the most and least important factors, respectively; there was significant heterogeneity among respondents in different subgroups in terms of choosing between *communication*, *trust*, *expertise*, *attitude*, and *literacy*, reinforcing the above logit model results. Among patients with different numbers of hospitalizations in the last year, there was a preference heterogeneity for *communication*, *trust*, *ability*, and *literacy*. Among those in various departments and different age groups, there was a preference heterogeneity for *attitude*, *communication*, *trust*, and *literacy*. Among those with different monthly family incomes, there was a preference heterogeneity for *communication*, *trust*, *expertise*, and *literacy*.

Concerning the number of hospitalizations in the last year, respondents hospitalized three or more times, as compared with the other two subgroups, perceived that *communication* more strongly affected their medical decision-making participation, and gave the lowest score to *ability* (those not hospitalized and those hospitalized once or twice gave higher scores to *ability*). Furthermore, respondents with a monthly family income of 10,000–20,000 CNY (1.428–2.856 USD), attached more importance to *expertise* and less to *communication* compared with the other subgroups; the opposite was true for those with a monthly family income under 10,000 CNY. Among different departments, compared with the other subgroups, respondents in the surgery department attached more importance to *attitude*, followed by those in the otolaryngology department. Among different age groups, respondents over 56 years old believed that *attitudes* more strongly affected decision-making participation, followed by those aged 46–55 and 36–45. Furthermore, respondents aged 26–35 gave the highest score to *communication*, whereas those aged more than 56 gave it the lowest score.

## 4. Discussion

In daily life, as individuals we usually weigh considerations of different standards using our intuition. However, when it comes to complicated or high-stakes decision-making, such as that regarding healthcare, it is critical and prudent to make better decisions by constructing the question and determining multiple standards [76]. Patient medical decision-making is a typical multi-criteria decision-making, as patient-centered care is based on patient preferences. Through statistical analyses, we can quantify, weigh allocation of, and rank patient-related clinical and nursing factors and identify the most/least popular option [77]. However, data with quantified and choice-based inpatient preferences are limited. No research has studied the ranking of factors’ importance in the context of patients’ medical decision-making participation and whether they are identical among respondents with different characteristics in China. Using the BWS approach, this study reduced the variance of scale and ranked the factors more precisely. Below, the influencing factors will be discussed from three perspectives: the patient, medical, and social environment.

From the patients’ perspective, patients’ trust in physicians, their ability to bear the disease burden, the influence of the people around them, their ability to participate, and their awareness of their illness, were the main factors associated with medical decision-making participation. Among the 13 factors, “patients’ trust in physicians” was the most important. Consensus on the influence of patients’ trust on decision-making participation is lacking. Some scholars suggest that patients’ trust is a special form of interpersonal trust and those with greater trust in physicians show better compliance, obtain better health outcomes, and participate more actively in decision-making [78,79]. For instance, Kraetschmer et al. [65] found that patients who prefer SDM might show medium-to-high trust levels. Peek et al. [80] revealed that physicians’ SDM behaviors can boost patients’ trust. Meanwhile, some studies show that patients’ trust negatively influences decision-making behaviors [81,82,83]. Patients’ trust in physicians makes it easier for both parties to reach a decision-making consensus because the former follows the latter’s advice. Therefore, patients are less likely to participate in decision-making. Moreover, patients’ trust in physicians significantly influences medical decision-making participation. Second, consistent with our findings, Wu et al. [84] found that in complex medical decision-making scenarios, the participation of people (especially relatives) around patients strongly influences patients. Some argue that the type and amount of medical information absorbed by patients impact their willingness and attitudes toward decision-making participation [85]; the better patients understand their diseases, or the better they can bear the disease burden, the more likely they are to actively adopt and engage in treatments [86,87]. In addition, in order to reduce exposure to COVID-19 (the data were collected during the COVID-19 outbreak, from 1 July–30 September 2020), many patients choose to minimise access to health care facilities, thereby reducing the in-person dissemination of information. As a result, the ability to locate, understand and use online health resources (i.e., e-health literacy) becomes more important during a pandemic. And patients with high levels of education and economic power tend to have higher levels of e-health literacy. Regarding heterogeneity, the MXL model revealed that inpatients with different demographic characteristics differed in their perception of the importance of trust and literacy. There was heterogeneity in the choice preferences of respondents in different departments, monthly family income, number of hospitalizations in the last year, and age groups. Consistent with Zhao et al. [88] but opposite to Kother et al. [15], inpatients’ trust scores increased with age. Furthermore, patients with different numbers of hospitalizations had different views on the importance of ability; for respondents hospitalized three or more times in the last year, ability had little influence on decision-making participation behaviors. This may be because they were already quite familiar with their diseases and had relevant knowledge, and they and/or their relatives already had strong participation abilities and therefore valued other factors such as the physicians’ communication ability.

From the physician’s perspective, professional expertise, attitude, communication ability, and consultation time duration were the most important factors, ranking among the top five. Among these four, only professional expertise was related to medicine. This finding was consistent with Zhang et al. [89] Moreover, patient participation reflected the degree to which patients and physicians interacted and depended on the exchange of emotions and information. Health workers usually cannot balance their time between treating and soothing patients especially during the COVID-19 pandemic [90,91]. Compared with outpatients, inpatients’ diseases are more severe, making it difficult to fully meet recovery expectations. Additionally, as some physicians lack training in interpersonal skills, they can face communication problems. They struggle to respond to patients’ emotional needs or provide necessary information [92], making patients participate less in decision-making and discuss treatment plans with relatives or friends instead. Therefore, health workers should develop communication abilities and improve service attitudes, while continuously strengthening professional expertise. Communication skill courses in hospitals should urge physicians to use language that is easy to understand to avoid confusion and a sense of distance resulting from miscommunication [93]. Regarding heterogeneity, the results indicated that respondents in different departments, with different monthly family incomes, with different numbers of hospitalizations in the last year, and in different age groups perceived the importance of communication differently. There was significant heterogeneity in the perception of the importance of attitudes among patients of different ages; there was heterogeneity in the perception of the importance of expertise among patients with different monthly family incomes. Respondents with a monthly family income of under 10,000 CNY valued physicians’ communication ability instead of professional expertise. Therefore, they were more concerned about feelings during physician interactions. Among the age groups, participants aged ≥56 years, followed by those aged 46–55 and those aged 36–45, viewed attitude as the most important factor; those aged 26–35 gave the highest score to communication, whereas those aged ≥56 gave it the lowest score, which was consistent with Li et al. [29]. From the physicians’ perspective, more patience and better communication attitudes are needed. Young patients are often more anxious about diseases, ready to communicate, and quick-witted; thus, they attach more importance to physicians’ communication abilities [29]. For patients in different departments, those in the surgery department gave the lowest score for communication. The reason for this may be that inpatients in the surgery department generally have more severe conditions and rely more on physicians. Moreover, this department is highly specialized, and physicians are generally busier, making them unable to communicate in-depth with patients.

For the social environment, hospital rules and regulations, medical laws and regulations, and the medical environment were correlated with patients’ medical decision-making participation; however, it ranked lower than the above factors, probably because patients were less familiar with laws, regulations, and rules than the factors related to physicians and themselves and were more inclined to ignore these factors. However, patients can better participate in medical decision-making by defending their rights, such as the right to informed consent, right to autonomy, and freedom of choice, if they know more about regulations related to medical decision-making [94]. Mohammed [95] discovered that most patients were not sufficiently aware of their rights. Furthermore, medical teams sometimes do not adequately brief patients on treatment regimens, making patients play a less important decision-making role. As patients become more concerned about and familiar with diseases, they are increasingly aware of the need to protect their rights in medical contexts, and legislation is required to protect their participation [94]. In the context of the COVID-19 pandemic, the Chinese government has introduced a series of documents and policies to promote national vaccination rates. Some studies have shown that although COVID-19 has hindered progress in SDM research, the choice of vaccination has increased public awareness of personal decision-making autonomy and the need to discuss the pros and cons with physicians before making medical decisions, which is a positive development. [96] Therefore, the government should develop supporting policies to clarify physicians’ responsibilities, improve hospitals’ codes of conduct, and increase patients’ awareness of their rights. Additionally, our study verified that individual differences were correlated with participation behavior. Thus, when communicating with patients about diseases and conditions, physicians should consider patients’ medical history, age, disease severity, and family economic situation to interact better with them and provide differentiated services with patient-centeredness at the core.

Our study enriched the use of the BWS method in healthcare in a Chinese context. The BWS approach, compared with traditional ranking approaches, has the advantages of less cognitive load, easier choice tasks, smaller sample size, and complete ranking, as well as a reduced influence of personal response styles [71]. Therefore, as a favorable new approach for eliciting abundant preference-related information, it should be widely used in future healthcare research.

### Limitations and Future Studies

This study has several limitations. First, to reduce the cognitive load on respondents, only 13 major attributes were included, which may lead to omissions. Future research could use bibliometric methods, such as using software like CiteSpace, to supplement attributes included in the survey. Second, Case 1 BWS is a relatively simple approach that does not assess trade-offs among different preference levels. Future studies could use Case 2 or 3 BWS to set up different attributes and levels, as well as their combinations, to obtain targeted findings. Third, this study was conducted in the more economically developed Zhejiang Province, China. The findings may not be generalizable to other regions. Convenience sampling was used, and consequently, the sample was not representative. Further research is needed on the heterogeneity of patients’ preferences for participating in decision-making and how to increase active and effective participation. Additionally, this study can be more innovative and fun. In other words, future research can explore the relationship between patient needs and physician needs and whether patients have participated in SDM interactions. Lastly, this study was not targeted, with respondents being inpatients with varied characteristics; thus, future research can target a specific group, such as patients with a certain disease.

## 5. Conclusions

Patient participation, regardless of the level, was strongly correlated with patients’ rights. Patient participation in clinical decision-making is necessary to establish treatment plans that address patients’ rights. It embodies respect for autonomy and dignity and can simultaneously reduce health providers’ decision-making errors [97]. From the inpatient perspective in the context of patient-centered medicine and adopting the BWS method, this study revealed the relative importance of factors affecting inpatient preference for participation in medical decision-making and quantified the preference heterogeneity among different groups. Generally, respondents viewed “patient trust in physicians,” “physician expertise,” and “physician attitudes” as the three most important factors, but the decision-making participation preferences of specific disease groups and their heterogeneity are yet to be examined.

## Figures and Tables

**Figure 1 healthcare-11-00323-f001:**
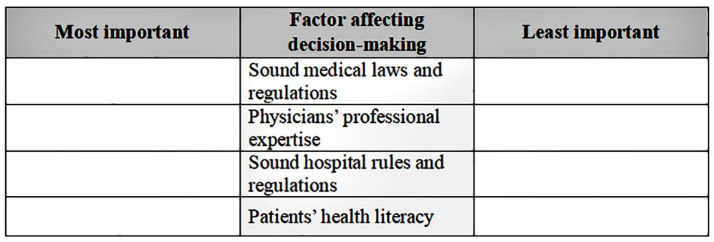
An example of the best-worst scaling questions.

**Figure 2 healthcare-11-00323-f002:**
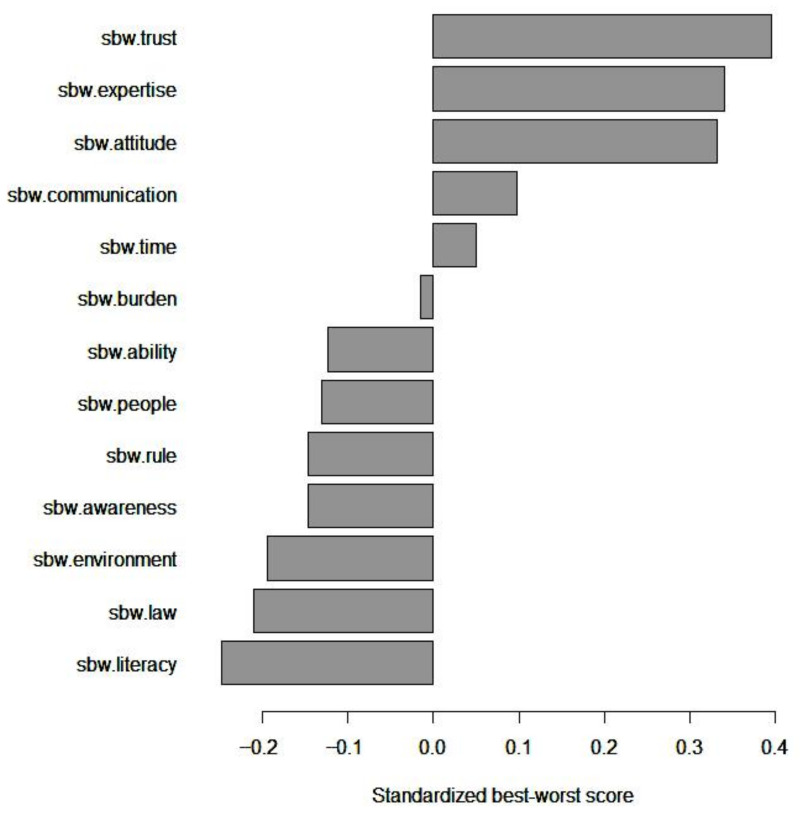
Bar plot of mean standardized best-worst scores.

**Figure 3 healthcare-11-00323-f003:**
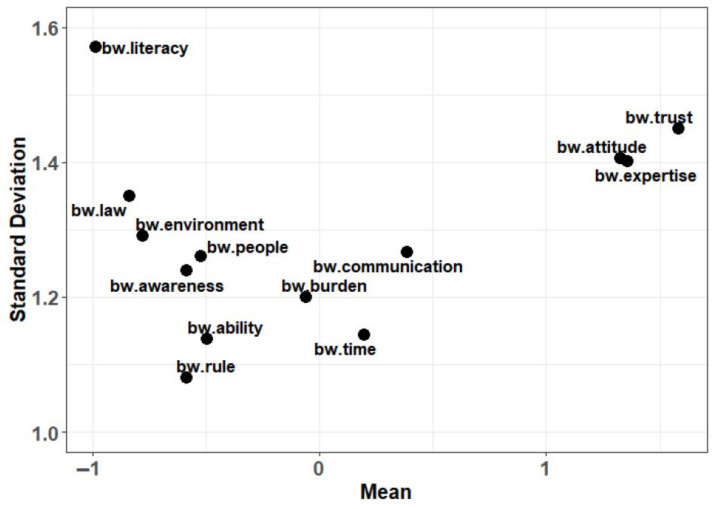
Plot of the mean and standard deviation of the best-worst score for each attribute (BW, best-worst).

**Figure 4 healthcare-11-00323-f004:**
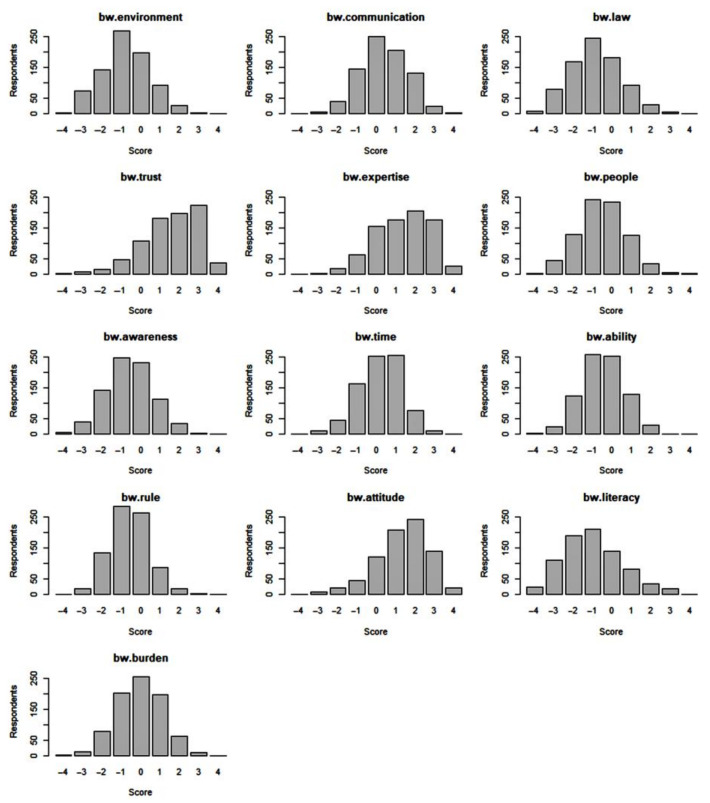
Empirical distribution of individual best-worst scores/bar plots of best-worst scores (BW, best-worst).

**Table 1 healthcare-11-00323-t001:** List of factors and descriptions.

	Factor	Abbreviation	Description
1	Sound medical laws and regulations	Law	The government has comprehensive laws and administrative regulations and a complete legal system to protect patients’ rights [58].
2	Sound hospital rules and regulations	Rule	The hospital the patient visits has clear and complete rules and regulations that govern health workers’ behaviors and clarify codes of conduct and reward and punishment mechanisms [59].
3	Medical environment	Environment	The medical environment should positively influence patients, that is, it should be convenient, comfortable, and patient-centered, and help patients’ recovery, satisfy their needs, and ease their pain [60].
4	Influence of the surrounding people	People	Comments by people around patients (e.g., families, friends, and colleagues) on the hospital or the physician, as well as successful cases of other people participating in decision-making [61].
5	Physicians’ attitudes	Attitude	The physician provides humane care, has good peer relationships, adheres to work ethics, and is passionate, sincere, calm, and careful [18].
6	Physicians’ professional expertise	Expertise	The physician shows a high level of clinical expertise and skills and is capable of achieving patients’ goals in terms of treatment [62].
7	Physicians’ communication ability	Communication	The physician can deliver necessary information with correct, accurate, and plain language, show humaneness and compassion, and listen to the patient [20].
8	Consultation time duration	Time	The physician has enough time for examination, diagnosis, treatment, and patient communication regarding illness and treatment [17].
9	Patients’ health literacy	Literacy	Patients can obtain and understand health information and use it to maintain or improve their well-being [63].
10	Patients’ awareness of their illness	Awareness	Patients can identify and understand their illnesses correctly and accurately [64].
11	Patients’ trust in physicians	Trust	Patients and physicians trust and respect each other; patients believe that physicians will do their best with regard to treatment [65].
12	Patients’ ability to participate	Ability	Patients can obtain medical information related to disease, treatment, and recovery before consultation, communicate or collaborate with physicians during the interaction, and have adequate ability to make decisions and protect their rights [27].
13	Patients’ ability to bear the disease burden	Burden	Patients can bear health or economic burdens related to pain, disability, and premature death resulting from their diseases [66].

**Table 2 healthcare-11-00323-t002:** Experimental design.

Factor	Choice Task (CT)												
	CT1	CT2	CT3	CT4	CT5	CT6	CT7	CT8	CT9	CT10	CT11	CT12	CT13
Environment	0	0	1	0	1	0	0	0	0	1	0	1	0
Communication	1	0	0	1	0	0	0	0	0	0	1	1	0
Law	0	0	1	1	0	1	0	1	0	0	0	0	0
Trust	0	1	1	0	0	0	0	0	0	0	1	0	1
Expertise	1	0	0	0	0	0	0	1	0	1	0	0	1
People	1	1	0	0	1	1	0	0	0	0	0	0	0
Awareness	1	0	1	0	0	0	1	0	1	0	0	0	0
Time	0	0	0	0	0	1	1	0	0	1	1	0	0
Ability	0	0	0	0	0	1	0	0	1	0	0	1	1
Rule	0	0	0	0	1	0	0	1	1	0	1	0	0
Attitude	0	1	0	1	0	0	0	0	1	1	0	0	0
Literacy	0	1	0	0	0	0	1	1	0	0	0	1	0
Burden	0	0	0	1	1	0	1	0	0	0	0	0	1

**Table 3 healthcare-11-00323-t003:** Descriptive statistics of the sample (*n* = 814).

Characteristics	Category	Frequency (*n*)	Composition Ratio (%)	Characteristic	Category	Frequency (*n*)	Composition Ratio (%)
Monthly family income (CNY)	<10,000	310	38.1	Number of hospitalizations in the last year	0	525	64.5
	10,00020,000	304	37.3		1~2	226	27.8
	>20,000	200	24.6		≥3	63	7.7
Age (y)	≤25	77	9.5	Department of hospitalization	Internal medicine	176	21.6
	26~35	204	25.1		Surgery	378	46.4
	36~45	186	22.9		Gynecology	86	10.6
	46~55	165	20.3		Otorhinolaryngology	82	10.1
	≥56	182	22.4		Other	92	11.3
Gender	Male	404	49.6	Academic degree	Middle school and below	467	57.4
	Female	410	50.4		College	145	17.8
					Undergraduate	164	20.1
					Masters or above	38	4.7

**Table 4 healthcare-11-00323-t004:** The results of the best-worst scaling survey.

	B	W	BW Score	Mean BW Score	Mean Std. BW Score	Scaled BW Score	Std. Scaled BW Score	Rank
Environment	594	1227	−633	−0.778	−0.194	0.696	0.279	11
Communication	966	650	316	0.388	0.097	1.219	0.489	4
Law	404	1,087	−683	−0.839	−0.210	0.610	0.245	12
Trust	1534	247	1,287	1.581	0.395	2.492	1	1
Expertise	1595	489	1,106	1.359	0.340	1.806	0.725	2
People	517	943	−426	−0.523	−0.131	0.740	0.297	8
Awareness	534	1,012	−478	−0.587	−0.147	0.726	0.291	10
Time	898	737	161	0.198	0.049	1.104	0.443	5
Ability	597	1000	−403	−0.495	−0.124	0.773	0.310	7
Rule	346	821	−475	−0.584	−0.146	0.649	0.260	9
Attitude	1393	313	1,080	1.327	0.332	2.110	0.847	3
Literacy	481	1285	−804	−0.988	−0.247	0.612	0.246	13
Burden	723	771	−48	−0.059	−0.015	0.968	0.389	6

B, best score; BW, best-worst; std., standardized; W, worst score.

**Table 5 healthcare-11-00323-t005:** Results of the conditional logit regression and mixed logit regression analyses.

	Conditional Logit Model				Mixed Logit Model			
	B	SE	*z*-Value	SP	B	SE	*z*-Value	SP
Trust	1.339	0.038	35.57 ***	0.1579	1.404	0.041	34.15 ***	0.1644
Expertise	1.205	0.037	32.64 ***	0.1382	1.250	0.035	35.42 ***	0.1409
Attitude	1.196	0.037	32.20 ***	0.1369	1.237	0.039	31.48 ***	0.1392
Communication	0.712	0.036	19.82 ***	0.0843	0.723	0.034	21.4 ***	0.0832
Time	0.604	0.036	16.93 ***	0.0757	0.612	0.035	17.35 ***	0.0744
Burden	0.465	0.035	13.12 ***	0.0659	0.469	0.036	12.94 ***	0.0646
Ability	0.256	0.035	7.25 ***	0.0535	0.258	0.035	7.29 ***	0.0523
People	0.242	0.036	6.80 ***	0.0527	0.245	0.035	6.91 ***	0.0516
Rule	0.218	0.036	6.10 ***	0.0515	0.221	0.039	5.74 ***	0.0504
Awareness	0.210	0.035	5.93 ***	0.0511	0.211	0.036	5.81 ***	0.0499
Environment	0.103	0.035	2.93 **	0.0459	0.104	0.034	3.05 **	0.0448
Law	0.086	0.036	2.41 *	0.0451	0.086	0.036	2.42 *	0.0440
Literacy	Reference			0.0414	Reference			0.0404

B, coefficient; SE, standard error; SP, share of preferences for the 13 items. * *p* < 0.05; ** *p* < 0.01; *** *p* < 0.001.

**Table 6 healthcare-11-00323-t006:** Best-worst scores of subgroups with different inpatient departments, monthly family incomes, number of hospitalizations in the last year, and ages (results of the one-way multivariate analysis of variance).

		Environment	Communication	Law	Trust	Expertise	People	Awareness	Time	Ability	Rule	Attitude	Literacy	Burden
Number of hospitalizations in the last year (times)	0	−0.181	0.102	−0.217	0.366 ***	0.334	−0.123	−0.151	0.038	−0.126	−0.131	0.331	−0.234 *	−0.007
	*1 or 2*	−0.215	0.062	−0.198	0.485	0.371	−0.135	−0.132	0.060	−0.093	−0.179	0.321	−0.312	−0.034
	≥3	−0.234	0.183 *	−0.191	0.318 **	0.278	−0.179	−0.167	0.111	−0.214 *	−0.151	0.373	−0.119 **	−0.008
	F-value	1.377	3.769 *	0.365	10.223 ***	1.940	0.890	0.447	2.063	4.573 *	2.540	0.544	6.813 **	0.665
Monthly family income (CNY)	*<10,000*	−0.186	0.131	−0.186	0.347	0.290	−0.112	−0.139	0.052	−0.151	−0.148	0.338	−0.219	−0.018
	10,000~20,000	−0.194	0.057 *	−0.220	0.438 **	0.371 *	−0.126	−0.163	0.067	−0.108	−0.122	0.326	−0.297 *	−0.029
	>20,000	−0.209	0.106	−0.230	0.406	0.369 *	−0.168	−0.135	0.018	−0.106	−0.179	0.331	−0.215	0.011
	F-value	0.316	4.324 *	1.261	4.980 **	5.018 **	1.943	0.658	1.868	2.268	2.721	0.093	3.962 *	1.101
Department	Internal medicine	−0.183	0.097	−0.185	0.401	0.288	−0.139	−0.158	0.068	−0.141	−0.149	0.315	−0.229	0.014
	*Surgery*	−0.189	0.058	−0.214	0.454	0.342	−0.120	−0.157	0.038	−0.107	−0.152	0.373	−0.296	−0.030
	Gynecology	−0.224	0.169	−0.250	0.294 **	0.352	−0.128	−0.076	−0.006	−0.134	−0.122	0.334	−0.154 *	−0.055
	Otolaryngology	−0.195	0.125	−0.171	0.265 **	0.369	−0.131	−0.156	0.101	−0.113	−0.131	0.217 **	−0.171	−0.009
	Others	−0.209	0.166 *	−0.236	0.353	0.391	−0.163	−0.141	0.068	−0.163	−0.150	0.294	−0.234	0.025
	F-value	0.304	3.812 **	0.983	7.388 ***	1.620	0.389	1.328	1.910	0.998	0.288	3.934 **	3.639 **	1.443
Age (years)	≤25	−0.175	0.107	−0.169	0.331 **	0.292	−0.127	−0.166	−0.010	−0.130	−0.097	0.302	−0.156 ***	−0.003
	26 to 35	−0.168	0.170 **	−0.200	0.346 ***	0.374	−0.162	−0.129	0.048	−0.154	−0.147	0.275 ***	−0.221 **	−0.032
	36 to 45	−0.200	0.077	−0.203	0.359 **	0.332	−0.100	−0.133	0.052	−0.106	−0.152	0.311 *	−0.199 **	−0.038
	46 to 55	−0.214	0.076	−0.230	0.412	0.342	−0.152	−0.155	0.080	−0.103	−0.147	0.336	−0.255	0.008
	*≥56*	−0.209	0.051	−0.227	0.500	0.327	−0.111	−0.166	0.045	−0.124	−0.158	0.426	−0.357	0.003
	F-value	0.663	4.162 **	0.612	6.067 ***	0.921	1.309	0.540	1.321	0.997	0.734	5.025 **	5.694 ***	0.849
Mean in the difference of the total		−0.194	0.097	−0.210	0.395	0.340	−0.131	−0.147	0.049	−0.124	−0.146	0.332	−0.247	−0.015

Demographic subgroups were compared with “1 or 2”,“<10,000”, “surgery”, and “≥56,” respectively, which are all italicized. * *p* < 0.05; ** *p* < 0.01; *** *p* < 0.001.

## Data Availability

The raw data supporting the conclusions of this article will be made available by the corresponding author upon request.

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
