# Peer review of "Best-Worst Scaling Survey of Inpatients’ Preferences in Medical Decision-Making Participation in China"

_healthcare, 2023, doi:10.3390/healthcare11030323_

Round 1
Reviewer 1 Report
The idea presented in the work is interesting but there are some lack as well the presentation issue in the manuscript. The following are the suggestions which need to be incorporated into the manuscript to improve its presentation.
The lines of the introduction section needs to be updated with some independent lines.
The introduction section should be updated by adding the objective and motivation related to application of given theorems.
Expand the introduction/literature review section so that readers can get motivation.
In this context, the following publications should be added to the references:
the following publications should be added to the references:
Discuss the major finding in a separate paragraph/section with respect to the existing study.
The similarity index should be reduced by adding new references.
There should be some studies for future research in this direction.
1) Kaya Pezük Åžeyda, ÅžENEL GÜZÄ°DE - 2021,An Application of Soft Multisets to a Decision-Making Problem Concerning Side Effects of COVID-19 Vaccines, Journal of new theory. 2021.
After this minor revisions, I can accept this manuscript. In addition, I would like to re-evaluate these edits for me to read in the new revision article. As a result, the admission status will be decided.
Author Response
Point-by-point response.
Reviewer 1:
1 The lines of the introduction section needs to be updated with some independent lines.
Response: Thank you very much for your valuable suggestions. We have revised and added to the Introduction, modified the Introduction structure, and divided it into Section 1.1. Thematic Background and Section 1.2. Methodological Background (see pages 1 and 2).
2 The introduction section should be updated by adding the objective and motivation related to application of given theorems.
Response: Thank you for pointing this out. We have clarified the objectives and motivations in the Introduction section, with details in the revised text (see page 2, lines 63-72, 87-88; page 3, lines 120-141).
3 Expand the introduction/literature review section so that readers can get motivation.In this context, the following publications should be added to the references:
Response: Thank you to the reviewers for this valuable suggestion, which we very much agree with. Recent research findings related to the topic have now been added to reduce the similarity index and enrich the literature review section. We also hope this will enhance the reader's motivation (see, for example, lines 44-54, 63-87, and 109-115).
4 Discuss the major finding in a separate paragraph/section with respect to the existing study.
Response: Thank you very much for your suggestion, we have revised the structure of the introduction, and collated and added a separate paragraph to summarize the main findings of the existing research. This has been detailed in the revised Introduction section (see pages 1 and 2, Sections 1.1. Thematic Background and 1.2. Methodological Background).
5 The similarity index should be reduced by adding new references.
There should be some studies for future research in this direction.
1) Kaya Pezük Åžeyda, ÅžENEL GÜZÄ°DE - 2021,An Application of Soft Multisets to a Decision-Making Problem Concerning Side Effects of COVID-19 Vaccines, Journal of new theory. 2021.
Response: Your suggestions and tips are much appreciated, and we have added your suggested citations and a number of other more recent literature reviews relevant to this study (see, for example, see lines 73-87).

Reviewer 2 Report
The proposed work aimed to assess inpatients’ preferences for participating in medical decision-making and determine the factors’ rankings in order of importance and whether they vary for respondents with different characteristics. The subject is interesting and a valuable contribution to the literature. Despite the notion that the work is well-described and provides valuable findings, there are a few aspects that should be taken into consideration before publishing.
1. To begin with, there are a few grammatical mistakes that need to be fixed.
2. There are a few acronyms that are not defined in the first place.
3. Though the introduction is articulate and crisp, I would suggest adding the most recent literature that supports the research questions of the present study and also the chosen methodology. This would highlight the importance of the proposed need.
4. As the authors have chosen a multicriteria decision-making technique i.e., BWM which is also reported as a sensitive approach with varying weights allocation and may affect the final decision of the investigation, how would you justify the reliability of the selected approach with the proposed work.
5. The reported attributes mentioned in table 1 based on the previous literature? If so, please provide references.
6. Discussion is presented strongly however not well backed up by previous literature in the multicriteria decision-making context. Please revise.
Author Response
Point-by-point response.
Reviewer 2:
1 To begin with, there are a few grammatical mistakes that need to be fixed.
Response: Thank you very much for your suggestion, we have double checked the grammar in the article and made the necessary corrections.
- There are a few acronyms that are not defined in the first place.
Response: Thank you very much for your suggestion, we have checked the article and ensured that all acronyms are defined.
- Though the introduction is articulate and crisp, I would suggest adding the most recent literature that supports the research questions of the present study and also the chosen methodology. This would highlight the importance of the proposed need.
Response: Thank you very much for your valuable suggestions. We have revised and added citations to the Introduction and added the latest literature to enrich the literature review section (see pages 1 and 2, Sections 1.1. Thematic Background and 1.2. Methodological Background; lines 44-54, 70-87, and 109-115).
- As the authors have chosen a multicriteria decision-making technique i.e., BWM which is also reported as a sensitive approach with varying weights allocation and may affect the final decision of the investigation, how would you justify the reliability of the selected approach with the proposed work.
Response: Thank you very much for the advice. Here is our explanation: In the present study, BWS was used as an approach for data collection based on the random utility theory—that is, assessing respondents’ preferences for different options by analyzing their choices among various preference questions, instead of using it as an approach for decision-making. We have elaborated on and explained this choice in the Introduction and Section 2.1 (see lines 151-174). As a multi-criteria decision-making technique, BWS could indeed involve weight allocation for various factors since each factor is evaluated with an item as a criterion, but we used it as a method of data collection and, specifically, used the BWS Case 1 which is designed to estimate the general rank of each item in a certain object list without a level structure and without the need of weight allocation. Unlike Case 1 that we used, Cases 2 and 3 present attributes and their levels, and therefore involve the issue of weight allocation (as described in Section 2.1). Besides, the questionnaire design and attribute selection in the study were reliable owing to the extensive literature review, rigorous expert panel discussions, focus group interviews, and strict quality control during questionnaire distribution (see Sections 2.2 and 2.4).
- The reported attributes mentioned in table 1 based on the previous literature? If so, please provide references.
Response: Thank you very much for your suggestion. We have added references to the definitions of the 13 attributes in Table 1 (see updated Table 1).
- Discussion is presented strongly however not well backed up by previous literature in the multi criteria decision-making context. Please revise.
Response: Thank you very much for your advice. We have added content relevant to multi-criteria decision-making. However, it is worth noting that, consistent with your advice, BWS was used in the study as an approach for data collection based on the random utility theory—that is, assessing respondents’ preferences for different options by analyzing their choices among various preference questions, instead of using it as an approach for decision-making. We have elaborated and explained this in the Introduction and Section 2.1 (see lines 99-109, and lines 151-174). In multi-criteria decision-making, BWS is regarded as a variant of the paired comparison method, and further research on this may be warranted.

Reviewer 3 Report
Aim of the study not clearly expressed
conclusion does not correspond entirely to the aim
add USD in brackets after Yuan
Author Response
Point-by-point response.
Reviewer 3:
1 Aim of the study not clearly expressed
Response: Thank you very much for your valuable suggestions. In the Introduction section, we have added relevant and up-to-date literature citations and rewritten the purpose of the study (see lines 133-141).
2 conclusion does not correspond entirely to the aim
Response: Thank you very much for your valuable suggestions. We have revised and improved the research aims and findings to make them correspond, (see lines 133-141, and 535-540).
3 add USD in brackets after Yuan
Response: Thank you for your valuable suggestions. We have added the equivalent in USD after Yuan and replaced the irregular RMB with CNY.

Reviewer 4 Report
Dear Authors,
Although I think it is interesting to know what patients in China need to participate in SDM, I have some concerns.
-Why would a best-worst case scenario method be better than just applying a prioritization, as in 1-10?
- Please elaborate on why not all completed questionnaires were included in the study.
-Table 3 is difficult to read.
-In Table 3, if I add all age ranges then it amounts to 533; where are all other 300 people?
-Gender, education level are descriptive data that are missing in table 3.
-The novelty and interest of the study can be improved by linking the needs of the patients to e.g. 1) the needs of the medical doctors or 2) whether patients were actually in a SDM conversation or 3) how trust can be generated (e.g. via qualitative interviews of patients)
-The population of the survey is now quite general. A more specific group would be more suited.
Author Response
Point-by-point response.
Reviewer 4:
-1 Why would a best-worst case scenario method be better than just applying a prioritization, as in 1-10?
Response: Thank you very much for the question. The most common types of rating scales are the Likert scale and 1-10 rating scale, but rating scales like 1-10 are susceptible to acquiescence bias (agreement bias), social desirability bias (tendency to lie), and extreme response bias which can produce biased results. Second, the results of 1-10 rating scales cannot show each item’s level of importance. Lastly, different rating scales are used in different countries, producing results that are not favorable for international comparison.
On the other hand, in BWS Case 1, different subsets of items are constructed from the list, and each subset is presented as a choice set to the respondents, who are asked to choose the best (or the most important) and worst (or least important) items. The task is repeated many times until all subsets are calculated. Compared to other ranking methods, BWS can minimize scale variance with the processes of evaluation, ranking, and selection, and is easier for respondents to operate and understand, facilitating fast reactions and consistent opinions. Also, with repeated questions, each item’s level of importance can be fully ranked, producing relatively more accurate and reliable rankings.
Many scholars have compared BWS with rating scales like 1-10, and results have shown that BWS can significantly reduce the three biases found in rating scales, increase visibility, and have an edge in the calculation of each item’s importance level (references cited below). Additionally, we have added descriptions of this method’s advantages in the Methods section (see Section 2.1, lines 151-163).
Jaeger T F. Categorical data analysis: Away from ANOVAs (transformation or not) and towards logit mixed models[J]. Journal of memory and language, 2008, 59(4): 434-446.
Adamsen J M, Rundle-Thiele S, Whitty J A. Best-Worst scaling... reflections on presentation, analysis, and lessons learnt from case 3 BWS experiments. Market & Social Research, 2013, 21(1).
Hein K A, Jaeger S R, Carr B T, et al. Comparison of five common acceptance and preference methods. Food quality and preference, 2008, 19(7): 651-661.
Cohen I L. Criterion-related validity of the PDD Behavior Inventory. Journal of Autism and developmental Disorders, 2003, 33(1): 47-53.
- 2 Please elaborate on why not all completed questionnaires were included in the study.
Response: Thank you very much for your suggestion. We have explained in Section 2.4 of the article the criteria for excluding some completed questionnaires—considered to be invalid (see lines 263-266).
-3 Table 3 is difficult to read. In Table 3, if I add all age ranges then it amounts to 533; where are all other 300 people?
Response: Thank you very much for your valuable suggestions, this was an oversight on our part and we are very sorry to have caused you any confusion. We have now amended Table 3.
-4 Gender, education level are descriptive data that are missing in table 3.
Response: Thank you very much for your valuable suggestions. Gender and education were not initially presented in Table 3 because subsequent analyses found no statistical significance in these two classifications for subgroup analysis. We have added descriptive statistics for gender and education to Table 3 and have added relevant text descriptions in Section 3.1 in the revised text (see lines 266, 328-333).
-5 The novelty and interest of the study can be improved by linking the needs of the patients to e.g. 1) the needs of the medical doctors or 2) whether patients were actually in a SDM conversation or 3) how trust can be generated (e.g. via qualitative interviews of patients)
-6 The population of the survey is now quite general. A more specific group would be more suited.
Responses 5 and 6: Thank you very much for your valuable feedback, which is very inspired. We strongly agree with you that this is a limitation of our current research (which is acknowledged in the Limitations in the Discussion section) and indicated as a direction for future research (see lines 535-540).
